# The Effect of Primary Duodenogastric Bile Reflux on the Presence and Density of *Helicobacter pylori* and on Gastritis in Childhood

**DOI:** 10.3390/medicina55120775

**Published:** 2019-12-05

**Authors:** Mehmet Agin, Yusuf Kayar

**Affiliations:** 1Department of Pediatric, Division of Pediatric Gastroenterology, Hepatology and Nutrition, Van Education and Research Hospital, 65300 Van, Turkey; 2Department of Internal Medicine, Division of Gastroenterology and Hepatology, Van Education and Research Hospital, 65300 Van, Turkey; ykayar@yahoo.com

**Keywords:** duodenogastric bile reflux, gastritis, *Helicobacter pylori*, age

## Abstract

*Background and Objectives:* Although there are many studies that investigate the relationship between duodenogastric reflux (DGR) and *Helicobacter pylori* in adult patients, the reported data are contradictory. In addition, there are very few studies in the literature investigating the relationship between DGR and *H. pylori* in the pediatric age group. In the present study, we investigated the effect of primary DGR on *H. pylori* and gastritis. *Materials and Methods:* A total of 361 patients who were referred to the clinic of our hospital with dyspeptic complaints who had an upper gastrointestinal system endoscopy and a gastric biopsy were included in the study. *Results:* DGR was detected in 45 cases, and 316 cases that did not have DGR were considered as the control group. Comparisons were made between the DGR cases and the control group in terms of risk factors (age, gender), the presence and density of *H. pylori*, and the presence and severity of gastritis. The average age of the patients who were included in the study was 11.6 ± 4.6 years. A total of 128 (36%) of the cases were male and 233 (64%) were female. DGR was present in 45 (13%) of the cases. The average age of the patients with DGR was 13.9 ± 3.1 years, the average age of the control group was 11.3 ± 4.7, and there were statistically significant differences (*p* < 0.001). No significant differences were detected in terms of gender between DGR and the control group (*p* > 0.05). *H. pylori* (+) was detected in 29 (64%) of patients with DGR, and in 202 (64%) of the control group. No significant differences were detected between *H. pylori* prevalence (*p* = 0.947). Gastritis was detected in 37 (82%) of the patients with DGR, and in 245 (77%) of the control group (*p* = 0.476). No significant differences were detected between the presence and density of *H. pylori*, gastritis presence, severity and DGR (*p* > 0.05). *Conclusions:* The ages of patients with DGR were significantly higher than in the control group, and advanced age was shown to be a risk factor for primary DGR. It was found that the presence of DGR has no effect on the presence and severity of *H. pylori*. Given this situation, we consider it is important to eradicate *H. pylori* infection, especially in the case where *H. pylori* is present together with DGR.

## 1. Introduction

Duodenogastric reflux (DGR) is the passing of the duodenal contents from the duodenum into the stomach [1]. Although DGR is quite common after cholecystectomy, pyloroplasty and gastric surgery, it can also develop primarily because of pyloric insufficiency without any secondary causes [1,2]. It was shown in previous studies that there might be a small amount of DGR physiologically in the stomach after feeding and in fasting [2,3]. The most important factor in the re-escape of the duodenum contents into the stomach is the deterioration in pylori function [4]. Although gastric acid and pepsin are the primary gastric agents providing the basis for mucosal damage and esophageal symptoms in the esophagus, damage occurs in the stomach and esophagus mucosa as a result of bile reflux and in the pancreas and with small intestine secretions from the duodenum into the stomach. Watt et al. showed that bile reflux into the stomach was associated with gastric ulcers and gastritis in patients who had dysfunction in the pyloric sphincter [5]. However, it is known that clinical complaints are disproportionate to mucosal damage. It was shown experimentally that DGR induced gastritis is not always symptomatic. Although complaints like epigastric pain, pain in the retrosternal region, and vomiting bile may develop after DGR, the presence and severity of symptoms were not found to be proportional to the amount of bile in the reflux [6,7]. Although DGR does not have a gold standard diagnosis, the presence of ulcers, erosion, fragility, mucosal erythema, and an abundant bile pool in the stomach seen in an endoscopy (especially in patients who have abdominal pain), epigastric pain, nausea and vomiting are characteristics of DGR [8,9,10].

*Helicobacter pylori* is a spiral-shaped Gram-negative bacterium that colonizes the gastric mucosa. *H. pylori* is the most common cause of resistant bacterial infection in the world and is located in the gastric mucosa and causes gastritis, peptic ulcers, B cell lymphoma and gastric cancer [11,12]. Although DGR and *H. pylori* cause different histopathological changes, *H. pylori* disrupts the mucosal barrier, initiates inflammation and sets the stage for an ulcer. Although there have been many studies to investigate the relationship between *H. pylori* and DGR in the adult age group, the data are contradictory [13,14]. In some studies, it was argued that the mucus barrier had deteriorated in the gastric mucosa because of increased bile acid, and as a result of this, *H. pylori* colonization decreases [13]. There are also other studies reporting that bile reflux increases gastric pH, disrupts the microenvironment needed for the survival of *H. pylori*, and eradicates it by disrupting colonization [13,14]. However, it was reported in some studies that there is a positive relationship between *H. pylori* and DGR, and DGR increased the presence of *H. pylori* [15,16]. There are no studies in Turkey that investigate the relationship between primary DGR and *H. pylori* in the childhood age group. In this study, we aimed to determine the relationship between primary DGR with the presence and density of *H. pylori* and gastritis, and the risk factors that affect DGR development.

## 2. Material and Methods

### 2.1. Patient Selection

A total of 361 cases who presented to the Pediatric Gastroenterology Clinic of the University of Health Sciences Van Education and Research Hospital between October 2017 and December 2018 with dyspeptic complaints who had undergone an upper gastrointestinal(GI) system endoscopy and gastric biopsy were included in the study. DGR was detected in 45 cases, and the 316 non-DGR cases were used as the control group. The age, gender, history, family history, history of metabolic disease, presence of concomitant systemic disease, upper GI system endoscopy results, histopathology results of the biopsies taken from the gastric mucosa, whether they were receiving *H. pylori* eradication therapy and abdominal ultrasonography (USG) data of the patients were recorded. In our study, the age, gender, family history, past surgical and interventional procedures, and the presence of pancreatic, hepatobiliary system and GI system pathologies seen in the abdominal USG were evaluated to determine the DGR risk factors. Those who underwent prior *H. pylori* eradication treatments, prior surgery for gastritis, prior gastric surgery, prior bile and/or bile duct surgery, prior endoscopic retrograde cholangiopancreatography (ERCP) procedures, those who received steroid and non-steroid anti-inflammatory treatment, those with diabetes, and those who had systemic comorbid diseases such as hypertension were excluded from the study.

### 2.2. Endoscopic Evaluation

The endoscopies of the patients were carried out by using a EG530WR Endoscopy (Fujinon, Tokyo, Japan) Device in the Endoscopy Unit of the Van Education and Research Hospital. Oral and written consent was received from the families before the endoscopy. All patients fasted for 6 h before the endoscopy, and after local pharyngeal xylocaine anesthesia, endoscopic procedures were carried out after the patients were sedated with midazolam (0.1 mg/kg) and ketamine (1 mg/kg). During the endoscopy, the esophagus, cardia, fundus, corpus and antrum regions of the stomach, and the duodenum were examined in detail. During the endoscopic examination, gastritis findings such as hyperemia, fragility, edema, gastric and duodenal ulcers and erosions, masses, hemorrhage, hiatal hernia, strictures and stenosis in the inferior esophageal sphincter, bile pool, and duodenogastric bile reflux were noted. The endoscopic data of the patients were recorded.

### 2.3. Histopathologic Evaluation

The endoscopic corpus and antrum biopsies were sent to the pathology laboratory in 10% formaldehyde. After the routine tissue follow-up procedures, the tissue samples that were embedded in paraffin were cut into 5-micron-thick slices, stained with routine hematoxylin–eosin (H-E), and evaluated under a light microscope. Following a rapid urease test to confirm the diagnosis of *H. pylori*, the biopsies were stained with modified Giemsa to histopathologically assess *H. pylori* presence and density. We measured the presence and density of *H. pylori* in patient biopsies, and the presence and severity of gastritis in accordance with the modified Sydney classification. Biopsies were assessed for the intensity of mononuclear inflammatory cellular infiltrates, inflammatory activity (neutrophilic infiltrations), glandular atrophy, dysplasia, metaplasia, and reparative atypia. Moreover, cases were graded according to the Houston and updated Sydney system, where the grading was in accordance with the intensity of mononuclear inflammatory cellular infiltrates within the lamina propria: absent inflammation (Grade 0), mild inflammation (Grade 1), moderate inflammation (Grade 2), and severe inflammation (Grade 3). The presence of *H. pylori* was interpreted as 1 (+), 2 (+) or 3 (+), and the severity of gastritis was interpreted as mild, moderate or severe [17,18].

### 2.4. Ethical Board Approval

Verbal and written informed consent was obtained from all the subjects included in the study and from their parents. After the study was completed, the study results for each subject was reported to their parents. Ethics committee approval for the study was given by the Van Education and Research Hospital Clinical Research Ethics Committee (Van/Turkey, Approval number 01/13, approved on 5 May 2019).

### 2.5. Statistical Analysis

The results of our study were analyzed with the Statistical Package for Social Sciences v.19.0 (SPSS Armonk, NY, USA: IBM Corp.) program. The data that had continuous values were given as average (mean ± standard deviation (SD)), and the categorical data were given as frequency and percentage (*n*,%). The data were tested for compliance in terms of normal distribution with the Kolmogorov–Simirnov test, histograms and ± SD. The Mann–Whitney Utest, Shaphiro–Wilk test and Student *t*-test were used for the analysis of non-parametric data of the groups, and the Chi-Square Test was used to analyze the categorical data. *p* < 0.05 was considered to be statistically significant.

## 3. Results

A total of 361 patients were included in the present study. The average age of the cases was 11.6±4.6 years. A total of 128 (36%) of the cases were male, and 233 (64%) were female. DGR was detected in 45 cases (13%), and not detected in 316 patients. Five cases with DGR (11%), and 31 non-DGR cases (10%) had kinship between parents. There was no statistically significant difference between the two groups. None of the cases had a family history of DGR, metabolic disease, prior gastric surgery, prior bile and/or bile duct surgery or prior ERCP. In all cases, *H. pylori* prevalence was 64% (231/361). In the evaluation of *H. pylori* density, the results were 1 (+) in 112 patients, 2 (+) in 80 patients, and 3 (+) in 32 patients. Gastritis was detected in 78% (282/361) of patients. In the evaluation made for gastritis severity, the results were mild in 135 patients, moderate in 112 patients, and severe in 35 patients (Table 1).

Patients who had DGR were compared with the control group (without DGR) to evaluate the effect of DGR on the prevalence and density of *H. pylori* and the presence and severity of gastritis. The average age of the patients who had DGR was 13.9 ± 3.1 years, and the average age of the control group was 11.3 ± 4.7 years; it was determined that the difference was statistically significant ( *p*< 0.001). When the relationship between gender and DGR was examined, it was determined that 15 (33%) of the patients with DGR were male, the number of the males in the control group was 113 (36%), and the difference was not at a significant level (*p* > 0.05). When the relationship between DGR and *H. pylori* was examined, it was determined that 29 (64%) of the patients who had DGR were *H. pylori* (+), and 202 (64%) of the control group were *H. pylori* (+). In the group with DGR, *H. pylori* density was 1 (+) in 13 (29%) patients, 2 (+) in eight (18%) patients, and 3 (+) in eight (18%) patients; in the patients in the control group, the *H. pylori* density was 1 (+) in 99 (31%) patients, 2 (+) in 72 (23%) patients, and 3 (+) in 31 (10%) patients. No significant differences were detected between the two groups in terms of *H. pylori* prevalence or *H. pylori* density (*p* = 0.947, *p* = 0.244, respectively). When the relationship between DGR and gastritis was analyzed, it was determined that gastritis was detected in 37 (82%) of cases with DGR, and in 245 (77%) of the control group. In the group that had DGR, gastritis severity was mild in 19 (51%) patients, moderate in 13 (35%) patients, and severe in five (14%) patients. In the control group, gastritis severity was mild in 116 (47%) patients, moderate in 99 (40%) patients, and severe in 30 (12%) patients. No significant differences were detected in terms of presence and severity of gastritis between the two groups (*p* = 0.476, *p* = 0.829, respectively) (Table 2). Logistic regression analysis was performed on all patients. We found that duodenogastric bile reflux had no effect on the severity of gastritis in the gastric mucosa, but that the severity of gastritis increased by 15.4 times in the presence of *H. pylori*. The frequency of *H. pylori* and gastritis in the DGR and control (non-DGR cases) groups is shown in Figure 1.

## 4. Discussion

Although there are wide-scale studies that investigate the relationship between DGR and *H. pylori* gastritis due to secondary causes, there is ascarcity of studies in which the relation between primary DGR and *H. pylori* gastritis is analyzed [19]. In our study, we aimed to investigate the relationship between primary DGR development, demographic data and *H. pylori* gastritis. In addition, age, gender, kinship, family history, metabolic disease, surgical and interventional procedures in their histories and the presence of pancreatic, hepatobiliary system and GI system pathologies in abdominal USG were evaluated to determine DGR risk factors. There were no other risk factors associated with DGR other than age. It was demonstrated that DGR was higher at a significant level in the elderly; however, no significant correlation was detected between DGR development and gender, density and presence of *H. pylori*, and presence and severity of gastritis.

In this study, we demonstrated that DGR was significantly higher in older children. The mean age of children with DGR was 13.9 ± 3.1 years, while the mean age of children without DGR was 11.3 ± 4.7 years. We did not find any previous studies on this subject in the pediatric age group. Previous studies in the adult age group have reported that the course of primary DGR is bimodal in terms of age and is more common in the younger age group (age range: 21–30 years) but is also common in the 50–80 year age range [19,20]. In our study, we found that there was no significant relationship between DGR and gender and that it was seen equally in both genders. We did not find any studies investigating the relationship between gender and DGR in the child population. Although there is one study examining the relationship between DGR and gender in the adult population, there are studies showing that it is 1.5:1 higher in females:males [21]. Studies have reported that *H. pylori* gastritis is seen equally in both genders because of the same probability of being affected by *H. pylori* infection, but DGR may be more common in women due to functional pathophysiology [19]. Mercan et al. reported that there was no significant gender difference between groups with and without DGR [22]. The results obtained from our study support this finding.

In our study, we found that DGR did not affect either the presence or severity of gastritis. However, it is still unclear whether primary DGR causes histological changes in the gastric mucosa in children [23,24]. It has been reported that short-term reflux of the duodenal contents into the stomach during physiological events can rarely cause symptoms. DGR is due to excessive reflux of bile, pancreas secretions and intestinal secretions into the stomach. An increased bile reflux rate may lead to increased gastric mucosal damage [23]. Biliary reflux secondary to surgery has characteristic histopathological changes such as foveolar hyperplasia, edema, hyperplasia in the lamina propria, and mild inflammation [23]. Wang et al. demonstrated that mild chronic inflammation occurred after prolonged reflux in their study in rats [24]. Due to DGR exposure, gastric polyps are present in studies reporting severe mucosal inflammation and intestinal metaplasia [25]. Taskin et al. found no significant difference in chronic inflammation between patients with and without DGR [26]. This study supports our data.

In our study, no statistically significant difference was found between the groups with and without DGR in terms of *H. pylori* prevalence or *H. pylori* density. Although both DGR and *H. pylori* infection are thought to damage the gastric mucosa, the relationship between DGR and *H. pylori* remains controversial. Although DGR has been reported to reduce the prevalence of *H. pylori* in some studies, other studies have reported a synergistic effect between DGR and *H. pylori*, while others still reported no significant relationship between them. Itoh et al. reported that bile acids had anti-bacteriological activity against *H. pylori* in vitro [27]. In previous studies, it was reported that bile inhibits *H. pylori* colonization by disrupting the microenvironment necessary for *H. pylori* colonization via disruption of the mucosal barrier and by forming an alkaline medium in the stomach, and it is concluded that bile eradicates *H. pylori* by this mechanism [13]. Sobala et al. [28] reported a negative correlation between the presence of DGR and the prevalence of *H. pylori*. In contrast to these data, Adam et al. in their study including 1120 pediatric patients found the prevalence of DGR to be 8.2% and the prevalence of *H. pylori* to be 25%; they also reported that *H. pylori* and DGR exhibited a synergistic effect [29]. Similarly, Ladas et al. reported that *H. pylori* could increase the toxic effects of DGR in intact gastric mucosa and that DGR was more frequent in *H. pylori* (+) cases and induced chronic gastritis with its synergistic effect [16]. However, no correlation was found between *H. pylori* intensity and DGR severity in either study. In contrast to these studies, Tewari et al. found no difference in the prevalence of *H. pylori* between patients with non-ulcer dyspepsia and biliary reflux gastritis [30]. Matsuhisa et al. also found the prevalence of *H. pylori* to be 27.2% and found no relationship between the presence or absence of DGR and the prevalence of *H. pylori* [31]. Taskin et al. also found no difference in *H. pylori* prevalence between patients with and without DGR [26]. Recent studies support our findings. The reason for the different results obtained in these studies can be due to the small number of cases evaluated in these studies, the presence of gastric surgery history, and differences in the amount and duration of duodenal reflux contents. Moreover, since bilirubin, which is regurgitated into the stomach, must act at the appropriate concentration and time to inhibit *H. pylori* infection, the present differences may be influential in the differences in outcomes [32].

Although there are strong aspects of our study, there are also some limitations. The strong aspects include: (a) investigating the age, gender, family history, parental kinship, metabolic disease, comorbid systemic diseases, anatomical anomalies and pathologies in terms of risk factors that may cause DGR through imaging; (b) evaluation of DGR in connection with the control group to examine its effects on *H. pylori* infection and the gastric mucosa; and (c) histopathological evaluation and comparison of *H. pylori* density and gastritis severity. However, only the fasting bile reflux was evaluated in our study. The fact that postprandial bile reflux and fasting bile reflux were not evaluated, and that the duration and amount of bile reflux to the stomach were not measured, are among the important limitations of our study.

## 5. Conclusions

The ages of the patients with DGR were higher at a significant level compared to those of the control group, and it was demonstrated that advanced age is a risk factor for primary DGR. It was also demonstrated that there was no significant difference between the cases with DGR and the control group in terms of the presence and density of *H. pylori* and gastritis presence and severity. However, it is already known that histopathological changes occur due to gastic mucosa exposure to bile acid and bile reflux, as well as the damage to the gastric mucosa due to *H. pylori* infection. Considering this, we believe that it is important to eradicate *H. pylori* infection, especially in the presence of DGR.

## Figures and Tables

**Figure 1 medicina-55-00775-f001:**
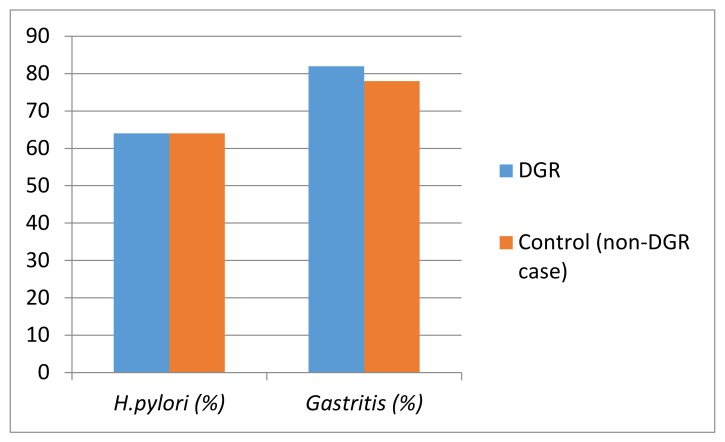
Comparison of the frequency of *H. pylori* and gastritis in the duodenogastric reflux (DGR) and control (non-DGR case) groups.

**Table 1 medicina-55-00775-t001:** The demographic, endoscopic and histopathologic data of the patients.

		Number	%
Gender	Male	128	36
	Female	233	64
Duodenogastric bile reflux	Yes	45	13
	No	316	87
*Helicobacter pylori*	Yes	231	64
	No	130	36
*Helicobacter pylori* density	1	112	31
	2	80	22
	3	39	11
Gastritis	Yes	282	78
	No	79	22
Gastritis severity	Mild	135	37
	Moderate	112	31
	Severe	35	10

**Table 2 medicina-55-00775-t002:** The relationship between DGR presence and demographic data, presence and density of *H. pylori*, and presence and severity of gastritis.

		Duodenogastric Bile Reflux		
		Yes	No	*p*
Age		13.9 ± 3.1	11.3 ± 4.7	<0.001
Gender	Male	15 (33%)	113 (36%)	0.750
	Female	30 (67%)	203 (64%)	
Presence of *Helicobacter pylori*	Yes	29 (64%)	202 (64%)	0.946
	No	16 (36%)	114 (36%)	
Density of *Helicobacter pylori*	1	13 (29%)	99 (31%)	0.244
	2	8 (18%)	72 (23%)	
	3	8 (18%)	31 (10%)	
Gastritis	Yes	37 (82%)	245 (78%)	0.476
	No	8 (18%)	71 (23%)	
Gastritis severity	Mild	19 (51%)	116 (47%)	0.829
	Moderate	13 (35%)	99 (40%)	
	Severe	5 (14%)	30 (12%)	

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
