# Peer review of "The Effect of Primary Duodenogastric Bile Reflux on the Presence and Density of Helicobacter pylori and on Gastritis in Childhood"

_medicina, 2019, doi:10.3390/medicina55120775_

Round 1

Reviewer 1 Report

There are some revisions as the following

Please define duodenogastric bile reflux. Is it the subjective observation determined by endoscopists or is there any objective endoscopic criteria in your study?. Different definitions may cause different results. Please specify what you define in your study ! In your patients, the diagnosis of H pylori infection is made based on histology. However, this diagnostic criteria is weak. Histology should be combined with rapid urease test to confirm Hp status. Please specify your diagnostic criteria for Hp infection. In your conclusion, you mentioned “histopathological changes occur due to exposure to bile acid of the gastric mucosa and bile reflux, as well as the damage on the gastric mucosa due to HP infection”. I suggest you should demonstrate this fact in your study. You have the data of the severity of gastritis, the status of Hp infection and the existence of duodenogastric reflux in each patient. Please demonstrate if combined Hp and duodenogastric reflux would affect gastritis severity or which of Hp or duodenogastric reflux affect gastritis severity more. This comparison has clinical importance. Please specify Please add pediatric patients in your title in order to clarify your study group ! In the conclusion of the abstract, you mentioned “ the reasons like the presence of gastric surgery history, the amount and duration of duodenal content in reflux, and the concentration of the bilirubin regurgitated to the stomach are effective in different results ”. However, your results have nothing to do with operation history, amount of duodenal reflux, bilirubin level regurgitated into the stomach. It is not appropriate to state that in your conclusion. Please state what’s your results instead of the results in other studies !

Author Response

REVÄ°EW 1.

Poınt 1-Please define duodenogastric bile reflux. Is it the subjective observation determined by endoscopists or is there any objective endoscopic criteria in your study?. Different definitions may cause different results. Please specify what you define in your study !

Answer1-Nowadays, no gold-standard diagnostic method for DGR exists, and no specific endoscopic or histopathological findings exist. A finding of bile content in the stomach prior to duodenum on endoscopy, as well as detecting gastric inflammation to different degrees during macroscopic and histopathologic examinations,could result in the clinician considering DGR. A hemorrhagic and vulnerable stomach wall, together with greenish stomach fluid,shows bile reflux. Even though DGR is absent of a gold standard diagnosis, detecting ulcers, mucosal erythema, abundant bile pool, erosion, and fragility in the stomach via endoscopy,particularly in patients reporting abdominal pain, vomiting epigastric pain, and nausea is characteristic of DGR.

Poınt 2- In your patients, the diagnosis of H pylori infection is made based on histology. However, this diagnostic criteria is weak. Histology should be combined with rapid urease test to confirm Hp status. Please specify your diagnostic criteria for Hp infection.

Answer-2 Following a rapid urease test to confirm the diagnosis of Helicobacter Pylori, the biopsies were stained with modified Giemsa to histopathologically assess HP presence and density, and were evaluated according to the updated Sydney classification.

Point 3- In your conclusion, you mentioned “histopathological changes occur due to exposure to bile acid of the gastric mucosa and bile reflux, as well as the damage on the gastric mucosa due to HP infection”. I suggest you should demonstrate this fact in your study. You have the data of the severity of gastritis, the status of Hp infection and the existence of duodenogastric reflux in each patient. Please demonstrate if combined Hp and duodenogastric reflux would affect gastritis severity or which of Hp or duodenogastric reflux affect gastritis severity more. This comparison has clinical importance.

Answer 3- Dear valued reviewer, you are correct about that. In this study, we employed regression analysis. We found that duodenogastric bile reflux had no effect on the severity of gastritis in the gastric mucosa, but that the severity of gastritis increased by 15.4 times in the presence of Helicobacter Pylori.

Point-4 Please specify Please add pediatric patients in your title in order to clarify your study group !

Answer 4- The Effect of Primary Duodenogastric Bile Reflux on the Presence and Density of Helicobacter Pylori, and on Gastritis in Childhood

Poınt-5.  In the conclusion of the abstract, you mentioned “ the reasons like the presence of gastric surgery history, the amount and duration of duodenal content in reflux, and the concentration of the bilirubin regurgitated to the stomach are effective in different results ”. However, your results have nothing to do with operation history, amount of duodenal reflux, bilirubin level regurgitated into the stomach. It is not appropriate to state that in your conclusion. Please state what’s your results instead of the results in other studies ! 

Answer 4.The conclusion section of the abstract is arranged in accordance with your suggestions as follows: ''The ages of patients with DGR were significantly higher than in the control group, and advanced age was shown to be a risk factor for primary DGR. The presence and density of DGR has no effect on the presence and severity of HP. Given this situation, we consider it is important to eradicate HP infection, especially in the case where HP is present together with DGR."

Reviewer 2 Report

The paper of Mehmet Agin and Yusuf Kayar investigates the “Effect of Primary Duodenogastric Bile Reflux on  the Presence and Density of Helicobacter Pylori and 3 on Gastritis”. The relationship between H. pylori and bile reflux gastritis is an interesting and poorly clarified field. Nevertheless, the design and the methods stimulate important doubts regarding the results of this study.

Objections are:

No mention is reported about the method used to detect duodenogastric reflux; obviously, this point represents an important limit for this study. Patients underwent esophagogstroduodenoscopy, but the finding of bile in the stomach is not a reliable tool for the diagnosis of bile reflux.

Author Response

REVÄ°EW-2

The paper of Mehmet Agin and Yusuf Kayar investigates the “Effect of Primary Duodenogastric Bile Reflux on  the Presence and Density of Helicobacter Pylori and 3 on Gastritis”. The relationship between H. pylori and bile reflux gastritis is an interesting and poorly clarified field. Nevertheless, the design and the methods stimulate important doubts regarding the results of this study.

Objections are:

No mention is reported about the method used to detect duodenogastric reflux; obviously, this point represents an important limit for this study. Patients underwent esophagogstroduodenoscopy, but the finding of bile in the stomach is not a reliable tool for the diagnosis of bile reflux. 

ANSWER- Nowadays, no gold-standard diagnostic method for DGR exists, and no specific endoscopic or histopathological findings exist. A finding of bile content in the stomach prior to duodenum on endoscopy, as well as detecting gastric inflammation to different degrees during macroscopic and histopathologic examinations, could result in the clinician considering DGR. A hemorrhagic and vulnerable stomach wall, together with greenish stomach fluid, shows bile reflux. Even though DGR is absent of a gold standard diagnosis, detecting ulcers, mucosal erythema, abundant bile pool, erosion, and fragility in the stomach via endoscopy, particularly in patients reporting abdominal pain, vomiting epigastric pain, and nausea is characteristic of DGR.

Reviewer 3 Report

Although the topic of the study is attractive, there are limitations to the methodological approach, presentation, and discussion. The study looks more like a mini-review than an original research.

Major comments

There is no description of the methods for detecting and measuring duodenal (bile). Also, there is no measurement of pH and no description for gastritis scoring (mild to severe). This point is important as the study is based on the relationship between bile reflux and the presence of HP and gastritis. The data presentation is not satisfactory. Using a chart should be a more appropriate way to illustrate the relationship between duodenogastric reflux and HP or gastritis. The Discussion section should emphasize the importance of their data, including interpretation.

Minor comment

The authors should have been updated with the most recent findings of experimental models, as well as clinical studies describing the oncogenic effect of gastroduodenal fluid in the mucosa, supporting it as a risk factor for malignancy.

Author Response

REVIEW-3

Although the topic of the study is attractive, there are limitations to the methodological approach, presentation, and discussion. The study looks more like a mini-review than an original research.

Major comments

POINT 1-There is no description of the methods for detecting and measuring duodenal (bile).

ANSWER 1:- Even though DGR is absent of a gold standard diagnosis, detecting ulcers, mucosal erythema, abundant bile pool, erosion, and fragility in the stomach via endoscopy, particularly in patients reporting abdominal pain, vomiting epigastric pain, and nausea is characteristic of DGR. Nowadays, no gold-standard diagnostic method for DGR exists, and no specific endoscopic or histopathological findings exist. A finding of bile content in the stomach prior to duodenum on endoscopy, as well as detecting gastric inflammation to different degrees during macroscopic and histopathologic examinations, could result in the clinician considering DGR. A hemorrhagic and vulnerable stomach wall, together with greenish stomach fluid, shows bile reflux (references: 8,9,10).

POINT 2- Also, there is no measurement of pH and no description for gastritis scoring (mild to severe). This point is important as the study is based on the relationship between bile reflux and the presence of HP and gastritis.

Answer 2- We measured the presence and density of Helicobacter Pylor in patient biopsies, and the presence and severity of Gastritis in accordance with the modified Sydney classification. Biopsies were assessed for the intensity of mononuclear inflammatory cellular infiltrates, inflammatory activity (neutrophilic infiltrations), glandular atrophy, dysplasia, and metaplasia, reparative atypia. Moreover, cases were graded according to the Houston and updated Sydney system, where the grading was in accordance with the intensity of mononuclear inflammatory cellular infiltrates within the lamina propria: absent inflammation (Grade 0), mild inflammation (Grade 1), moderate inflammation (Grade 2), and severe inflammation (Grade 3) (references:17,18).

POINT 3- The data presentation is not satisfactory. Using a chart should be a more appropriate way to illustrate the relationship between duodenogastric reflux and HP or gastritis.

Answer 3- As suggested by you, a graph showing the relationship between HP and Gastritis of DGR has been produced.

POINT 4- The Discussion section should emphasize the importance of their data, including interpretation.

Answer 4- As suggested by you, It is corrected.

Minor comment

POINT 5-The authors should have been updated with the most recent findings of experimental models, as well as clinical studies describing the oncogenic effect of gastroduodenal fluid in the mucosa, supporting it as a risk factor for malignancy.

ANSWER-5: One of the most recent studies in the literature has been added.

Round 2

Reviewer 1 Report

Dear authors,

Thanks for your revision and additional literature reviews. First, the definition of duodenogastric reflux was mentioned in your introduction part with some references and the Hp diagnosis was also defined specifically in your methods. In the results, you also demonstrated how either Hp or DGR affect the severity of gastritis as I suggested in the previous reply. The conclusion of your abstract were also revised based on your results instead of results from other study. These revisions make your manuscript complete and qualified for acceptance.

Author Response

Dear authors,

Thanks for your revision and additional literature reviews. First, the definition of duodenogastric reflux was mentioned in your introduction part with some references and the Hp diagnosis was also defined specifically in your methods. In the results, you also demonstrated how either Hp or DGR affect the severity of gastritis as I suggested in the previous reply. The conclusion of your abstract were also revised based on your results instead of results from other study. These revisions make your manuscript complete and qualified for acceptance.

ANSWER: Distinguished Arbitrator, first of all, we would like to thank you for your time and interest in our study.

Reviewer 3 Report

Lines 166-167: It is stated that “Logistic regression analysis was performed on patients and we found that gastritis increased 15.4 times in patients with DGR, but DGR had no effect on gastritis. A. we cannot see in figure 1 that gastritis increased 15.4 times with DGR vs control.Also, it is not demonstrated in the same figure that DGR had no effect on gastritis. Please, explain.

Axis y: Please add a title Explain in the figure legend what is the control (non-DGR case).

Lines 188-200: Rewrite the paragraph: The reader is interested to see the authors’ findings first, and then how these findings improve our knowledge.  (For example, the authors said: “DGR is significantly higher in older children”. We would like to know: what was the age range of these childer?; is this a novel finding according to previous studies? What did the previous studies refer about the incidence of DGR and older children’s age?)

By a similar way Lines 201-212 and 214-233 should summarized.

Author Response

Comments and Suggestions for Authors

POINT-1-Lines 166-167: It is stated that “Logistic regression analysis was performed on patients and we found that gastritis increased 15.4 times in patients with DGR, but DGR had no effect on gastritis. A. we cannot see in figure 1 that gastritis increased 15.4 times with DGR vs control.Also, it is not demonstrated in the same figure that DGR had no effect on gastritis. Please, explain.

ANSWER-1.

Dear valued reviewer, first of all, we would like to thank you for your time and interest in our study. 

You are right about the following;

We applied logistic regression analysis on patients. We found that DGR had no effect on the presence or severity of gastritis in the DGR group. In the presence of HP, we detected a 15.4-fold increase in both severity and frequency of gastritis in both groups. Since the prevalence of HP was very close in both the DGR and Control groups, we did not find any change in the frequency of gastritis.

POINT-2. Axis y: Please add a title Explain in the figure legend what is the control (non-DGR case).

ANSWER-2. As suggested by you, It is corrected.

POINT-3. Lines 188-200: Rewrite the paragraph: The reader is interested to see the authors’ findings first, and then how these findings improve our knowledge.  (For example, the authors said: “DGR is significantly higher in older children”. We would like to know: what was the age range of these childer?; is this a novel finding according to previous studies? What did the previous studies refer about the incidence of DGR and older children’s age?)

By a similar way Lines 201-212 and 214-233 should summarized.

ANSWER-3. As suggested by you, It is corrected.
